# WATCH-PR: Comparison of the Pulse Rate of a WATCH-Type Blood Pressure Monitor with the Pulse Rate of a Conventional Ambulatory Blood Pressure Monitor

**DOI:** 10.3390/bioengineering12050492

**Published:** 2025-05-05

**Authors:** Mathini Vaseekaran, Marcus Wiemer, Sven Kaese, Dennis Görlich, Jochen Hinkelbein, Gerrit Jansen, Alexander Samol

**Affiliations:** 1Department of Cardiology and Critical Care Medicine, Johannes Wesling University Hospital, 32429 Minden, Germany; marcus.wiemer@muehlenkreiskliniken.de (M.W.); sven.kaese@muehlenkreiskliniken.de (S.K.); alexander.samol@st-antonius-gronau.de (A.S.); 2Department of Anesthesiology, Intensive Care Medicine and Emergency Medicine, Johannes Wesling Hospital, University Hospital Ruhr-University Bochum, 32429 Minden, Germany; jochen.hinkelbein@muehlenkreiskliniken.de (J.H.); gerrit.jansen@muehlenkreiskliniken.de (G.J.); 3Institute of Biostatistics and Clinical Research, University Münster, 48149 Muenster, Germany; dennis.goerlich@ukmuenster.de; 4Department of Cardiology and Angiology, St. Antonius-Hospital Gronau GmbH, Möllenweg 22, 48599 Gronau, Germany

**Keywords:** pulse rate, smartwatch, monitoring, oscillometric measurement, atrial fibrillation, telemetric healthcare

## Abstract

Background: Monitoring pulse rate is fundamental to cardiovascular health management and early detection of rhythm disturbances. While oscillometric blood pressure measurement is well established and validated in clinical practice, its use for pulse rate monitoring, particularly via wrist-worn devices, remains largely unexplored. Objective: This study investigates whether a smartwatch that performs oscillometric blood pressure measurements at the wrist can also deliver reliable pulse rate readings using the same method. Methods: This study compared pulse rates recorded by the Omron HeartGuide smartwatch and conventional ambulatory blood pressure monitors in 50 patients over 24 h. Measurements were taken consecutively, and data were analyzed using intraclass correlation coefficients (ICCs) and Bland–Altman plots. Results: The study showed a high ICC of 0.971, indicating excellent agreement between devices. The average pulse rate difference was 1.5 bpm, with the Omron HeartGuide reporting slightly lower rates, especially in patients with atrial fibrillation. Conclusions: This study demonstrates that oscillometric pulse-rate monitoring at the wrist can achieve a high degree of accuracy, comparable to conventional upper-arm devices. Given that oscillometric smartwatches like the Omron HeartGuide are already used for blood pressure monitoring, the findings suggest that they may also be suitable for pulse rate measurement, potentially enhancing their role in telemetric healthcare, but further research is needed, particularly in patients with arrhythmias.

## 1. Introduction

Heart rate monitoring is a crucial aspect of cardiovascular health management, offering insights into an individual’s overall well-being. The pulse rate, the number of palpable pulse waves per minute, is a vital indicator of cardiovascular function and can reveal much about an individual’s health status [1,2,3]. It can signal the efficiency of the heart in pumping blood, the presence of any cardiovascular abnormalities, and the impact of various factors such as stress, physical activity, and sleep on heart health. A normal resting pulse rate for adults typically ranges from 60 to 80 beats per minute (bpm) [4]. Variations within this range can indicate different states of health, e.g., hyperthyroidism. An elevated pulse or heart rate has been linked to increased overall mortality in numerous studies [5,6,7,8].

Traditionally, pulse rate monitoring has been conducted in clinical settings using specialized equipment. However, with advancements in technology, smartwatches have become valuable tools for independent and point-in-time pulse rate monitoring [9,10,11,12]. Smartwatches equipped with pulse or heart rate sensors offer numerous benefits for health monitoring, especially in outpatient care settings where continuous monitoring can enhance disease management and early intervention [13,14,15,16].

Smartwatches equipped with photoplethysmography (PPG) sensors have long been used for pulse rate monitoring. Recently, there have been initial attempts to extend the use of PPG-based devices to blood pressure measurement; however, accuracy remains limited, especially under motion or arrhythmic conditions [17,18,19]. In contrast, oscillometric blood pressure measurement—widely regarded as the gold standard for non-invasive monitoring—has now been successfully implemented in some wearable devices, such as the Omron HeartGuide, using a cuff-based mechanism at the wrist.

Oscillometric methods appear to provide more reliable blood pressure readings than PPG-based approaches. However, it remains unknown whether this technology can also deliver accurate pulse rate measurements when applied via wearable devices. To date, no studies have systematically evaluated pulse rate measurement using oscillometric techniques in smartwatches [20,21]. Additionally, few investigations have compared pulse rate measurements between the radial artery (wrist) and brachial artery (upper arm), raising further questions about potential site-specific differences [22].

The aim of this study is to evaluate whether pulse rate values measured by the Omron HeartGuide smartwatch, using an oscillometric method at the wrist, are consistent and comparable to those measured by a conventional ambulatory blood pressure monitor at the upper arm. This investigation addresses a critical gap in the literature and explores the potential of oscillometric wrist-based monitoring as a reliable method for pulse rate assessment, especially in the context of telemetric and outpatient care.

The main scientific contributions of this work are as follows:-We present one of the first comparative studies analyzing pulse rate measurement using a cuff-based oscillometric smartwatch (Omron HeartGuide) against a conventional ambulatory blood pressure monitor.-We provide evidence for the accuracy of oscillometric pulse-rate monitoring at the wrist, a method that is underrepresented in the current literature.-We analyze the influence of atrial fibrillation and patient characteristics (e.g., BMI, age, wrist circumference) on pulse rate agreement, highlighting potential clinical implications.-We demonstrate the telemetric potential of oscillometric smartwatches for out-of-hospital pulse rate monitoring, contributing to the advancement of wearable health technologies.

The remainder of this paper is structured as follows: Section 2 describes the materials and methods, including the study population, the measurement devices, and the pulse rate measurement protocol. Section 3 presents the results of the study, including statistical comparisons and subgroup analyses. In Section 4, we discuss the main findings in the context of the existing literature, highlight clinical implications, and address the study’s limitations. Finally, Section 5 summarizes the conclusions and outlines future research directions.

## 2. Materials and Methods

This study was conducted in accordance with the Declaration of Helsinki and was approved by the Ethics Committee of the Medical Faculty of Ruhr University Bochum, file number 2020-647, date 26 June 2020. All participants provided informed consent prior to inclusion in the study. The present study is a subgroup analysis of date data derived from the “WATCH-BPM—Comparison of a WATCH-Type Blood Pressure Monitor with a Conventional Ambulatory Blood Pressure Monitor and Auscultatory Sphygmomanometry” study [23]. Fifty consecutive patients who underwent 24 h blood pressure monitoring for diagnostic purposes were equipped with an Omron HeartGuide watch during this period. These patients were instructed to use the Omron HeartGuide to measure their blood pressure and pulse rate one minute after the ambulatory blood pressure monitor had taken measurements on their upper arm. The recordings from both the ambulatory blood pressure monitor and the Omron HeartGuide were retrieved after 24 h. The measured pulse-rate values from both devices were then compared.

### 2.1. Study Population

After giving informed consent, 50 outpatients aged ≥18 years who were scheduled for a 24 h long-term blood pressure measurement in clinical practice were recruited consecutively at the cardiology department of Johannes Wesling University Hospital in Minden, University Hospital of Ruhr University Bochum, Germany, and were included in this study. Exclusion criteria were pregnancy and breastfeeding.

### 2.2. Pulse Rate Measurement Device

The Omron HeartGuide device (Omron Healthcare Co., Ltd. in Kyoto, Japan; https://www.omron-healthcare.de/produkte/heartguide (accessed on 25 March 2025)) is a patient-initiated automatic oscillometric device designed for measuring blood pressure and pulse rate at the wrist [21,23]. In our study, we utilized the Omron HeartGuide model HEM-6411T-MAE, selected for its advanced capabilities in oscillometric measurement at the time of the study initiation. It is important to position the wrist at heart level during blood pressure measurements to minimize hydrostatic effects [21,23]. The device features automatic inflation of the cuff via an electric pump and deflation through a mechanical valve. By analyzing the pulse wave detected during inflation with a specialized algorithm, the device determines systolic blood pressure (ranging from 60 to 230 mmHg), diastolic blood pressure (ranging from 40 to 160 mmHg), and pulse rate (ranging from 40 to 180 beats per minute) [24].

We selected the appropriate cuff size for wrist circumferences between 16.0 and 19.0 cm. Blood pressure and pulse rate data collected by the device were wirelessly transmitted via Bluetooth to a smartphone equipped with the Omron HeartAdvisor app (year 2020). The app displays all recorded values with corresponding dates and times, allowing for easy access and management. These data lists can be conveniently transferred to a personal computer (PC) through the app [25]. All information was securely stored in pseudonymized format on the PC in a MS Excel file.

The conventional ambulatory blood pressure monitor used in this study was the Spacelabs 90217A-2 (Spacelabs Healthcare GmbH (Company in Nürnberg in Germany), serial number 217A-015530). The device was calibrated according to the manufacturer’s specifications prior to study initiation, ensuring accurate and reliable measurements throughout the monitoring period.

### 2.3. Pulse Rate Measurement Protocol

The Omron HeartGuide (worn on the wrist) and the ambulatory blood pressure monitor (worn on the upper arm) were placed on the same arm. The ambulatory blood pressure monitor measured blood pressure and pulse rate automatically every 30 min during the day (6:00 a.m. to 10:00 p.m.) and every 60 min at night (10:00 p.m. to 6:00 a.m.). Patients were instructed on how to measure their blood pressure and pulse rate with the Omron HeartGuide and were asked to take these measurements one minute after the upper arm device had recorded its readings. They were also informed of the correct position to ensure accurate measurements with the Omron HeartGuide as shown in Figure 1.

The first measurement was supervised by qualified medical staff. All subsequent measurements were performed without supervision. For the initial measurement, participants were seated in a quiet room at a comfortable temperature, with their back supported, legs uncrossed, and the measurement arm supported so that the wrist was at heart level (this position is necessary for the Omron HeartGuide but not for the ambulatory blood pressure monitor). Blood pressure and pulse rate measurements were started after a five-minute resting period. Wrist circumference was then measured. Sixty seconds after the ambulatory blood pressure monitor’s measurements, blood pressure was measured using the Omron HeartGuide.

### 2.4. Statistical Analysis

The statistical analyses were conducted similarly to those in the blood pressure study [23]. In brief, statistical analyses were performed using SPSS version 28. The primary outcome measured was pulse rate. We calculated the 24 h average from the measurements as an aggregated measure for the primary outcome.

Pulse rate data from all measurement methods were summarized using means and standard deviations. We analyzed the associations between baseline variables (age, BMI, and wrist circumference) and pulse rate using Spearman’s rho correlation. To evaluate the reliability of the Omron HeartGuide, we used a Bland–Altman plot, comparing its measurements with those from the ambulatory blood pressure monitor.

For each patient, we plotted the difference between the Omron HeartGuide and ambulatory blood pressure monitor measurements against the mean of the two values to assess any systematic bias, such as consistent under- or overestimation of blood pressure. The Bland–Altman plots showed the mean difference as a solid horizontal line, along with the lower and upper 1.96 standard deviations of the mean difference. We used these plots to check if most measurements fell within the tolerance ranges, defined as the lower and upper 1.96 standard deviations of the mean difference. If so, the Omron HeartGuide was deemed sufficiently accurate compared to the standard measurement method.

To further analyze the association between pulse rate measurement methods, we calculated the two-way mixed intraclass correlation coefficient (ICC). The ICC measures the consistency or agreement of measurements made by different instruments or observers and is particularly suitable for assessing the reliability of repeated measurements.

The ICC values were interpreted according to Cicchetti and Koo and Li’s criteria: According to Cicchetti, ICC values less than 0.4 indicate poor reliability (“bad”), values between 0.4 and 0.59 indicate moderate reliability (“average”), values between 0.6 and 0.74 indicate good reliability (“good”), and values greater than 0.74 indicate excellent reliability (“very good”). According to Koo and Li, ICC values less than 0.5 indicate poor reliability (“bad”), values between 0.5 and 0.75 indicate moderate reliability (“average”), values between 0.75 and 0.9 indicate good reliability (“good”), and values greater than 0.90 indicate excellent reliability (“very good”).

We also compared non-aggregated data between the Omron HeartGuide and the ambulatory blood pressure monitor measurements by calculating differences. These differences were displayed as a scatter plot over time (24 h), and the general trend was assessed using LOESS regression. No inferential statistics were performed for this exploratory analysis.

To explore whether patient characteristics influenced discrepancies between the two measurement methods, we conducted Spearman’s rank-order correlation analyses between the absolute differences in pulse rate readings and continuous variables, including age, body mass index (BMI), and wrist circumference. The use of Spearman’s rho was chosen due to its robustness in detecting monotonic relationships without assuming normal distribution of the variables.

Furthermore, to assess the potential effect of clinical conditions and medication on device agreement, we generated scatter plots in which the mean pulse rate of both devices was plotted against the absolute measurement difference. These plots were stratified by the presence of atrial fibrillation, other arrhythmias, and beta-blocker therapy, thereby enabling a visual comparison of measurement bias across clinically relevant subgroups.

## 3. Results

The study involved 50 participants (male = 54%; age: 52.3 ± 14.5 years) from Western ethnic groups who were consecutively screened in our outpatient clinic for validation purposes. The mean wrist circumference was 17.6 ± 1.3 cm, the average BMI 29.3 ± 6.1 kg/m^2^. Table 1 shows the characteristics of the patients included.

A total of 968 measurements were taken using the smart device, with 811 paired measurements of ambulatory blood pressure monitor and Omron HeartGuide successfully obtained. The mean difference in pulse rate between the two devices was 1.5 ± 3.5 bpm, with a high ICC value of 0.971 (Table 2). The ICC according to Cicchetti and according to Koo and Li was “very good”.

If we look at the study divided into two patient groups, namely patients with and without atrial fibrillation, the results are as follows: In seven patients with atrial fibrillation, the mean pulse rate difference was 4.5 ± 5.2 bpm. In the 43 patients without atrial fibrillation, the mean pulse rate difference was 0.9 ± 2.9 bpm. To evaluate the relationship between device differences and variables such as age, BMI, and wrist circumference, Spearman correlation analyses were conducted. The results showed no significant correlation between these variables and the differences in pulse rate measurements. As shown in Figure 2, the Bland–Altman plot reveals a mean difference of 1.5 bpm between the two devices, with the majority of the data points falling within the ±1.96 standard deviation limits. These findings indicate good agreement between the devices, although some minor deviations were observed, particularly in measurements from patients with atrial fibrillation. The LOESS plot in Figure 3 demonstrates minimal fluctuation in pulse rate differences over the 24 h period, with no evident time-dependent patterns or trends. Recruitment details are provided in Table 3, and the impact of atrial fibrillation on pulse rate differences between the devices is illustrated in Figure 4. Participants with atrial fibrillation showed lower pulse rates on the Omron HeartGuide compared to the ambulatory blood pressure monitor, while no clear trend was observed in participants taking beta-blockers or with other forms of arrhythmia.

## 4. Discussion

Summing up, 809 paired measurements between an ambulatory blood pressure monitor and the Omron HeartGuide were taken. The mean pulse rate difference was 1.5 ± 3.5 bpm with a high ICC of 0.971, showing very good agreement. No correlation was found between device differences and age, BMI, or wrist circumference. Participants with atrial fibrillation had lower pulse rates on the Omron HeartGuide compared to Ambulatory blood pressure monitor.

Traditional pulse rate monitoring often requires clinical equipment and is unavailable when symptoms occur. Many patients are unable to palpate their own pulse, and data transfer to physicians frequently relies on manual documentation, which is prone to errors and delays. Wearable smart devices offer a promising solution by enabling automated, real-time data collection and transmission, potentially improving patient care. Validated data on their use in daily clinical practice are increasingly available. [1]. However, there is no investigation about smart devices that can measure the pulse rate oscillometrically. Additionally, to our knowledge, there are no comparative data specifically addressing oscillometric pulse measurement differences between radial (wrist) and brachial (upper arm) sites.

Studies have demonstrated the reliability of oscillometric methods for various clinical applications, such as ankle–brachial index (ABI) measurements. Kollias et al. (2011) validated oscillometric ABI measurements against Doppler, showing high agreement and clinical relevance for diagnosing peripheral arterial disease (PAD) [8]. Results from former studies indicate that the mean difference in blood pressure between the Omron HeartGuide and the ambulatory blood pressure monitor was acceptable in both office and out-of-office settings [21,23]. To our knowledge, we conducted the first study using the Omron HeartGuide device for pulse rate analysis.

To roughly answer the question of reproducibility of pulse rate values recorded by conventional ambulatory blood pressure monitors when compared with those obtained using the Omron HeartGuide, an ICC value of 0.971 is excellent and indicates a very strong correlation between both devices [26,27].

On average, the Omron HeartGuide in our study measured the pulse rate 1.5 bpm lower than the ambulatory blood pressure monitor. However, a deviation of 1.5 bpm would not be clinically relevant. Previous studies have indicated that PPG-based smartwatches tend to underestimate pulse rate measurements compared to reference methods or standards. This discrepancy varies based on the specific device and the conditions of use, such as the level of physical activity or clinical settings [19,28]. In another study, where an oscillometric blood pressure monitor on the upper arm was compared with manual blood pressure measurement, the pulse rate measured on the upper arm at rest was on average 0.9 bpm higher than the manual pulse measurement [29]. However, it is unknown which artery was used for the manual pulse measurement. The most likely one is the radial artery. If that were the case, in most studies, including ours, the pulse rate measured distally would be minimally lower than proximally (at heart level or on the upper arm). Concurrently, there has been longstanding debate over whether radial blood pressure measurement is less accurate than upper arm measurement [30,31]. But, there are no comparative data specifically addressing oscillometric pulse measurement differences between radial and brachial sites. Our data, as well as the data from the recently mentioned study [29], show that, at least as measured oscillometrically, the measurement on the upper arm seems to be more accurate. Our study highlights differences in pulse rate measurements between radial (wrist) and brachial (upper arm) sites. Araujo-Moura et al. (2022) emphasized the importance of validating oscillometric devices across various anatomical sites, especially given the variability in measurement conditions [14]. The observed discrepancies in our data suggest that further standardization and cross-validation of algorithms for wrist and upper arm measurements are needed. This is particularly evident in the group of participants with atrial fibrillation. With a regular pulse or frequent ventricular extrasystoles that occur close to the previous contraction, a peripheral pulse deficit could be the cause of the described results, but in the case of atrial fibrillation there could be another possible reason, which is described below.

As of recent estimates in 2019, approximately 59.7 million people globally are affected by atrial fibrillation [32]. This number represents a significant increase from previous decades and highlights the growing prevalence of atrial fibrillation across the world [33]. Atrial fibrillation poses significant health risks and is a common reason for increased hospitalizations and healthcare [33]. Therefore, accurate monitoring is especially important for atrial fibrillation. Figure 4 shows that, particularly in patients with atrial fibrillation, the pulse rate measured by the Omron HeartGuide was lower than that measured by the ambulatory blood pressure monitor. Even in absolute numbers as shown in the results, the Omron HeartGuide performs worse in patients with atrial fibrillation. Studies show PPG-based smartwatches are less accurate in measuring pulse rate during physical activity than at rest [19,34]. Physical activity and atrial fibrillation both increase heart rates and strain the cardiovascular system. Clinical comparisons of PPG sensors and electrocardiograms during atrial fibrillation show mixed results [18,26]. Another study showed that diagnostic accuracy for detecting atrial fibrillation of the electronic blood pressure monitor and especially the handheld electrocardiogram exceeded that of radial pulse palpation [35].

Patients with atrial fibrillation present unique challenges for oscillometric pulse-rate monitoring due to irregular ventricular rates. Xie et al. (2020) found that higher ventricular rates (>100 bpm) significantly impair the accuracy of oscillometric measurements [36]. Similarly, Su and Guo (2022) reported that rapid and irregular pulse intervals in AF lead to inconsistent readings, necessitating algorithmic adaptations for wearable devices [22]. These findings align with our observations of lower pulse rate measurements in AF patients using the Omron HeartGuide. Recent studies confirm these limitations: Zhou et al. (2024) highlighted the challenge in differentiating AF from sinus rhythm using oscillometric monitors; Zhao et al. (2024) evaluated algorithm-guided PPG monitoring and observed limitations in quantifying AF burden; and Sibomana et al. (2025) concluded in a meta-analysis that PPG-based smartwatches are generally less accurate than ECG patches in detecting AF. Our findings align with this body of literature and underscore the need for algorithmic adaptation when using oscillometric wrist devices in patients with arrhythmia [12,16,20]. It has been recommended that with an irregular heartbeat, the pulse should be measured over one minute to prevent fluctuations due to the irregular pulse interval [35,37]. This means that the different measurements between the upper arm and the wrist could simply result from different measurement times and not be related to the length distance between the two measurement sites. This could explain the previous findings from the studies, namely that the pulse rate is measured lower in cases of atrial fibrillation.

The absence of significant correlations between device differences and variables such as age, BMI, and wrist circumference suggests that the performance of the devices is not influenced by these individual factors. This finding supports the robustness of the devices in providing consistent pulse rate measurements across a diverse population. However, it is important to note that the sample size and variability within these factors might limit the detection of subtle associations. Future studies with larger and more diverse cohorts could provide further insights into the potential impact of these variables.

Since the Omron HeartGuide provides accurate pulse rate measurements in addition to blood pressure, it could be used to monitor patients’ vital parameters. In the long term, this could relieve nurses from manually measuring vital parameters multiple times a day, as the values could be transmitted from the watch. In addition to inpatient use, deployment in the outpatient sector is also possible, as the study with ambulatory patient shows. During the 24 h measurement, only the first measurement was supervised; the remaining measurements were performed without supervision, so it cannot be guaranteed that they were carried out correctly. Measurements with the Omron HeartGuide must be manually initiated. Therefore, patients using this smart device for diagnostic purposes need to be highly compliant and perform the measurements correctly or the results may be inaccurate. Improved ease of use for the Omron HeartGuide would be desirable in the future. Currently, automatic night measurements are not possible with the Omron HeartGuide. Simply waking up can increase the pulse rate and thus not provide true nighttime resting values. The inflation of both cuffs is noticeable to the patient; therefore, the pressure from both cuffs alone could lead to an increased pulse rate that might not be present during an unnoticed measurement.

Nevertheless, the Omron HeartGuide delivered very good results. However, it should be noted that this watch tends to display lower pulse rate values compared to measurements taken at the upper arm, particularly in patients with atrial fibrillation. This difference is also observed, albeit to a lesser extent, in patients without rhythm disturbances. These findings suggest that pulse rate measurements at the wrist may systematically differ from those taken at the upper arm.

Future studies should focus on improving the usability of wearable devices like the Omron HeartGuide, particularly for patients with AF. Enhanced algorithms tailored for arrhythmias and automated nocturnal measurements could further increase the clinical utility of such devices. As Su and Guo (2022) suggested, the current limitations of oscillometric methods in irregular rhythms must be addressed to ensure broader applicability in telemetric healthcare [22].

### Limitations

One limitation of this study is the use of different devices for pulse rate measurements at the upper arm and wrist. However, this approach is unavoidable, as devices designed for oscillometric blood pressure measurements at the upper arm cannot be used at the wrist and vice versa. Importantly, both devices used in this study rely on the same measurement method—oscillometry—for pulse rate detection. This ensures consistency in the underlying measurement principle, which is critical for a meaningful comparison.

Nevertheless, it is acknowledged that differences in device-specific algorithms and hardware may introduce variability. This limitation does not undermine the primary objective of the study but should be taken into account when interpreting the results. Future studies could focus on further standardizing measurement methods across devices to minimize potential discrepancies.

Our study involved a relatively small patient sample with 50 patients. A larger patient cohort would enhance the conclusiveness of the results. Additionally, the study was neither randomized nor blinded. The measurements were conducted by a single individual, preventing any comparisons or assessments of measurement quality across different operators. In 24 h, an average of only 16 pulse rate values per patient were recorded using the smart device, which could be attributed to patient compliance issues.

## 5. Conclusions

This study demonstrates that the Omron HeartGuide, a smartwatch utilizing oscillometric technology, provides pulse rate measurements at the wrist that are highly consistent with those obtained from a conventional upper-arm ambulatory blood pressure monitor. With a high intraclass correlation coefficient (ICC) of 0.971 and a small mean difference of 1.5 bpm, the results confirm the device’s potential for accurate pulse rate assessment in everyday settings. These findings are particularly relevant for patients who require regular monitoring, as the device allows for cuff-based measurements of both blood pressure and pulse rate using a single wearable tool. This could support remote monitoring, reduce reliance on manual documentation, and enhance patient autonomy. However, deviations in patients with atrial fibrillation indicate a need for improved algorithmic accuracy under irregular rhythm conditions. Future studies should focus on validating these findings in larger and more diverse populations, assessing automated measurement features, and optimizing usability—particularly for patients with arrhythmias or limited compliance in outpatient settings.

## Figures and Tables

**Figure 1 bioengineering-12-00492-f001:**
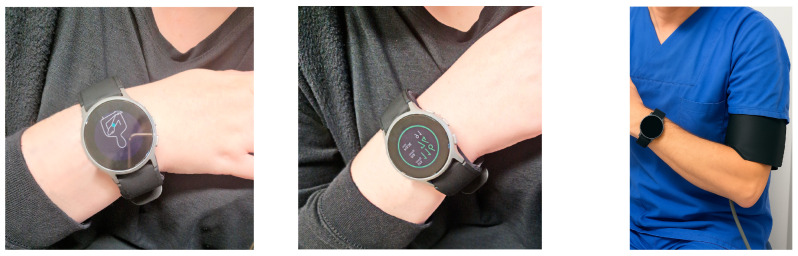
Measurement with Omron HeartGuide: The device is positioned at heart level in accordance to the instructions on the display (**left**); after the measurement, blood pressure values are displayed (**middle**); photograph with both devices, the Omron HeartGuide (worn on the wrist) and the ambulatory blood pressure (worn on the upper arm) (**right**).

**Figure 2 bioengineering-12-00492-f002:**
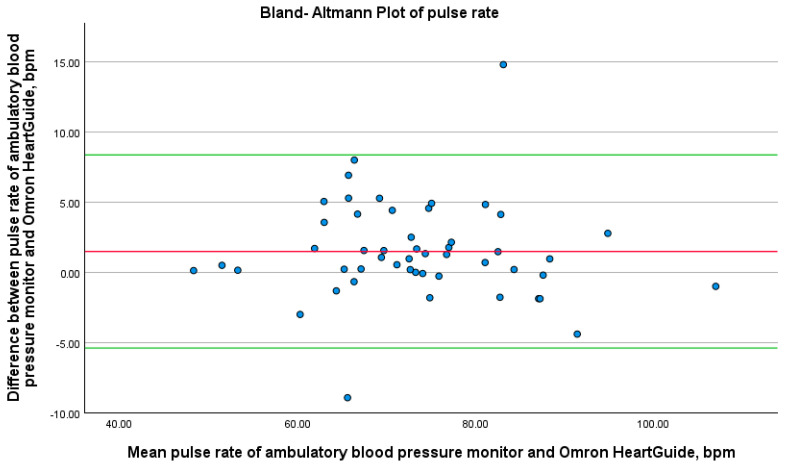
Bland–Altman plot of pulse rate. Bland–Altman plots for the pulse rate differences between the readings of the ambulatory blood pressure monitor and Omron HeartGuide. Thick red solid line = mean difference; green line = ±1.96 standard deviations of the mean difference.

**Figure 3 bioengineering-12-00492-f003:**
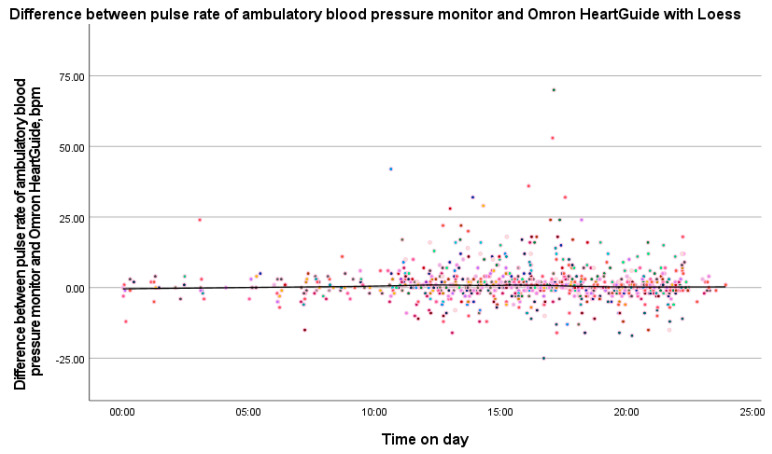
Difference between the pulse rate of ambulatory blood pressure monitor and Omron HeartGuide with LOESS. LOESS plot with the pulse rate difference of ambulatory blood pressure monitor and Omron HeartGuide throughout the day. Each colored dot represents one measurement from an individual participant; colors are used to visually distinguish patients from each other.

**Figure 4 bioengineering-12-00492-f004:**
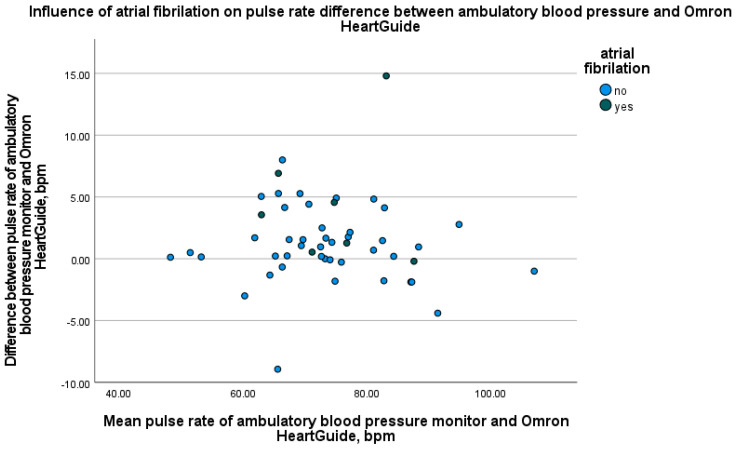
Influence of atrial fibrillation on PR difference between ambulatory blood pressure monitor and Omron HeartGuide. Light blue dots = participants without atrial fibrillation, dark blue dots = participants with atrial fibrillation.

**Table 1 bioengineering-12-00492-t001:** Characteristics of the study participants.

Age, y	52.3 ± 14.5
Men:women, n	27:23
Wrist circumference, cm	17.6 ± 1.3
Height, m	1.75 ± 0.1
Weight, kg	90.0 ± 20.9
Body mass index, kg/m^2^	29.3 ± 6.1

Data are expressed as the mean ± standard deviation or percentages or number.

**Table 2 bioengineering-12-00492-t002:** Comparison of pulse rate measured by Omron HeartGuide and ambulatory blood pressure monitor.

	Omron HeartGuide	Ambulatory Blood Pressure Monitor	Difference (Ambulatory Blood Pressure Monitor—Omron HeartGuide)	Correlation Coefficient
Pulse rate, bpm	72.8 ± 11.3	74.3 ± 11.1	1.5 ± 3.5	0.971

**Table 3 bioengineering-12-00492-t003:** Recruitment details: n = 50.

Variable	Number of Participants (n)
Smokers	10
Cardiovascular disease	38
Participants with atrial fibrillation	7
Participants with arrhythmia other than atrial fibrillation	2
Participants under beta blocker treatment	24

## Data Availability

Data are available from corresponding author. All authors take responsibility for all aspects of the reliability and freedom from bias of the data presented and their discussed interpretation.

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
