# Peer review of "WATCH-PR: Comparison of the Pulse Rate of a WATCH-Type Blood Pressure Monitor with the Pulse Rate of a Conventional Ambulatory Blood Pressure Monitor"

_bioengineering, 2025, doi:10.3390/bioengineering12050492_

Round 1

Reviewer 1 Report

Comments and Suggestions for Authors

1) In the abstract, the problem domain has not been well articulated. You need to revise this section so as to answer the following questions:
    i) What are the specific problems that you sought to solve?
    ii) What are the current solutions to problems in (i) above?
    iii) What are the research gaps/weaknesses of the current solutions in (ii) above?
2) The research area and research problem (s) have not been well articulated in the Introduction section
3) Towards the end of Section 1, do the following:
   i) In point form, describe the main scientific contributions of this work. Ensure that novelty is reflected in each of these contributions.
   ii) After the contributions in (i) above, add a paragraph to detail how this paper is structured.
4) The quality of English used throughout this paper must be greatly enhanced. In addition, there are numerous grammatical mistakes that must be eliminated through proper proof-reading. For instance, the following sentences (among others) have mistakes that must be eliminated:
 "The This study was conducted in accordance with the Declaration of Helsinki..."
--Check on 'The This'
"Many studies showed that an increased pulse or heart rate in particular are significantly associated 43 with an increasing risk of all-cause mortality [4–7]."
-- Check on 'all-cause mortality'
"were interpreted according to Cicchetti and Koo & Li's criteria: According to Cicchetti, ICC values less than 0.4 indicate poor reliability (“bad”), values between 0.4 and 0.59 indicate moderate reliability (“avarage”), values between 0.6 and 0.74 indicate good reliability (“good”), and values greater than 0.74 indicate excellent reliability (“very 151 good”)."
-- This sentence seems incomplete.

-- Check on the spelling of 'avarage'
5) The title for sub-section 2.1 is 'Study population'. However, you have not provided the specific number of participants for this study.
6) The conclusion section should be one paragraph detailing the key findings based on the study objectives. In addition, you need to articulate the practical implications of the obtained results, as well as the future research scopes.
7) The list of references must be revised to include more of the articles published in  the year 2024 and 2025

Reviewer 2 Report

Comments and Suggestions for Authors

The paper evaluates the use of a smart watch to measure the pulse rate. The technique proposed has been compared with the traditional method of pulse rate monitoring. The paper is well written, the typical sections of a research paper have been well-considered, and the methodology has been described very clearly. The abstract and Conclusions sections are very clear.

The work is very nice but authors should review the presentation of the results, and they also should review the writing mode in several parts of the paper.

I encourage authors to follow these recommendations.

MAJOR QUESTIONS

1.- Please indicate a photograph where both devices, the Omron HeartGuide (worn on the wrist) and the ambulatory blood pressure (worn on the upper arm) can be seen.

2.- At the end of section "2.4. Statistical Analysis", please describe the last two paragraphs with  more rigor

3.- Figure 2. Include a legend with the different colors of the lines 

4.- Figure 3. A legend with the different colors should be included, or explain better the different colors that you are using

5.- Figure 4. Improve the legend, it is not clear

6.- Please, you should review the Discussion Section, you should focus on the results, sometimes this section seems the state of the art typical of the Introduction Section

MINOR QUESTIONS

At the end of section 2, correct "The This study"

Page 4, line 148, correct "were" at the beginning of the sentence

Page 4, line 178, correct "In", after ":".

Please review the paper to detect these minor errors

Reviewer 3 Report

Comments and Suggestions for Authors

This study compared the pulse rate measurement consistency between Omron heartguide smart watch and traditional ambulatory blood pressure monitor (ABPM) through prospective design. The topic has clinical value, especially in the context of the increasing popularity of wearable devices. The research design is rigorous, the statistical method is reasonable, and the data is clear, which provides new evidence for oscillation pulse rate monitoring. But it needs further improvement.

  1. It is necessary to clarify the specific model, manufacturer and calibration information of ABPM to ensure the comparability of evaluation equipment.
  2. The author analyzed that there were significant differences in patients with atrial fibrillation, which may lead to the attenuation of oscillatory signal or misjudgment of algorithm due to irregular rhythm, and relevant literatures should be cited for analysis.
  3. Suspected typographical errors in Table 3.
  4. The full text uses "pulse rate" and "heart rate" alternately, which should be unified.
  5. "Ambulatory blood pressure monitor" keeps abbreviation consistency.

Round 2

Reviewer 1 Report

Comments and Suggestions for Authors

The authors have addressed all my concerns.